# Effects of unilateral neck muscle vibration on standing postural orientation and spatial perception in healthy subjects based on stimulus duration and simultaneous stimulation of trunk muscles

**Masaki Hirosawa**[1,2], **Itaru Takehara**[3], **Yoshinao Moriyama**[1,2], **Kazu Amimoto**[4]*

1 Department of Physical Therapy, Tokyo Metropolitan Rehabilitation Hospital, Tokyo, Japan, 2 Department of Physical Therapy, Graduate School of Human Health Sciences, Tokyo Metropolitan University, Tokyo, Japan, 3 Department of Rehabilitation Medicine, Tokyo Metropolitan Rehabilitation Hospital, Tokyo, Japan, 4 Faculty of Human Health Sciences, Department of Physical Therapy, Tokyo Metropolitan University, Tokyo, Japan

* amimoto@tmu.ac.jp

**Data Availability Statement:** The data underlying the results presented in the study are available

## Abstract

Neck muscle vibration (NMV) influences proprioceptive sensations and modulates standing postural orientation and spatial perception. However, the effects of NMV in healthy participants would vary based on the influence of stimulus duration and combination with trunk muscle vibration. Therefore, this study with a cross-over design clarified these effects. Twenty-four healthy participants (mean age, 25.7±3.7 years) were enrolled. To assess standing postural orientation, standing center-of-pressure (COP) measurements were recorded on a COP platform, starting with closed eyes and then with open eyes. The mean mediolateral (ML) and anteroposterior (AP) position [mm] of COP and other parameters were calculated. To assess spatial perception, subjective straight ahead (SSA) measurements were recorded, wherein participants were instructed to point and project the position of the manubrium of sternum on the touch panel using their right index finger with their eyes closed. Measurements were taken before and after four conditions: no vibration (control), left NMV for 30 s, left NMV for 10 min, and left NMV and left lumbar back vibration for 10 min. Vibratory stimulation was performed with the eyes closed at 80 Hz. The measurements under the four conditions were conducted with random cross-over and 5-min resting period between the conditions. COP and SSA values were subtracted before and after each condition for standardized variation and compared. NMV combined with trunk muscle vibration for 10 min resulted in significant deviations of the ML-COP toward the stimulation side and AP-COP toward the anterior side compared to the control condition with closed eyes. SSA showed no significant differences. These findings suggest that NMV-induced nervous system modulation would be amplified by proprioceptive sensory input to trunk muscles. Therefore, this method could provide a new option for clinical trials on postural orientation using NMV. SSA based on proprioceptive sensation may not be biased without visual illusions.

from the figshare (https://doi.org/10.6084/m9.figshare.20235606.v1).

**Funding:** The authors received no specific funding for this work.

**Competing interests:** The authors have declared that no competing interests exist.

## Introduction

Postural orientation requires the integration of vestibular, visual, and somatosensory information. Perception of the body in space also enables postural control and body movements [1,2]. Focal muscle vibratory stimulation is a powerful stimulus that selectively activates the primary endings of muscle spindles [3,4] and is used to provide sensory input to proprioceptive receptors. Sensory input to muscle spindles produces the illusion of muscle lengthening [5]. The compensatory response of the central nervous system (CNS) to the illusion of focal vibratory stimulation of the postural muscles of the neck, trunk, and lower limbs is a tilted standing posture [6]. The proprioception system of the cervical muscles is directly related to vestibular and eye movements [5,7,8] and plays a vital role in the perception of the body in space and the localization of posture by detecting the position of the head relative to the trunk [9]. Vibratory stimulation of the neck muscles, known as neck muscle vibration (NMV), has been reported to modulate postural orientation and spatial perception [9,10].

The effect of NMV on postural orientation differs during and after stimulation [9]. The standing center of pressure (COP), an index of postural orientation, deviates in the opposite direction of the stimulated side during NMV in healthy participants [5,6,11]. However, previous studies on the immediate effects of NMV in healthy participants reported that the standing COP immediately deviated to the same side as the stimulus after 30 s of NMV [12] and that the standing COP did not immediately deviate after 5 min of NMV [13]. In patients who had undergone unilateral leg amputation and had a biased standing COP, 30 s of NMV to the amputated side resulted in a greater bias of standing COP to the amputated side after stimulation than in healthy participants [12]. In a previous study of patients with a stroke and biased standing COP, the COP was biased to the paralyzed side after 10 min of NMV stimulation [14]. Therefore, the immediate effect of NMV on the direction of COP deviation is unclear, as previous studies have shown varying results depending on the stimulation time and the participant's condition.

The subjective straight ahead (SSA), a measurement that indicates the error between the objective midline of the body and subjective midline of perception, has been used as an index of midline perception in the horizontal plane [15]. The SSA deviates toward the stimulus side during and after NMV, in both healthy and stroke populations [9,10]. However, there are no reports on the effects of NMV on SSA and COP measured simultaneously in healthy participants.

The effect of NMV is enhanced by increasing the amplitude [16], prolonging the stimulation duration [17], and repeated daily stimulation for 14 days [18]. However, the effect of NMV alone is known to be abolished by superficial sensory inputs such as light touch [19], suggesting that the impact of NMV on the modulation of sensory integration may be limited. The local vibratory stimulation of the neck and trunk deviate the standing COP to the same side as the stimulus [5]. Moreover, among the stimulated sites of the whole body, when the muscles of the upper trapezius and trunk's lumbar region were vibrated, the standing COP deviated to a similar direction [6]. In addition, the effect of neck and trunk muscle stimulation in SSA has not yet been clarified. Therefore, we hypothesized that the posterior effect of NMV on standing COP is amplified by combining NMV with vibratory stimulation of the trunk muscles.

Thus, this study aimed to clarify the effects of NMV on standing postural orientation and spatial perception in the same healthy participants, depending on differences in stimulation duration and simultaneous stimulation of trunk muscles.

## Materials and methods

### Study design and participants

This was an interventional study with a cross-over design. Approval was obtained from the Research Safety Ethics Review Committee of the Tokyo Metropolitan University Arakawa

Campus (approval number: 20021) and the Tokyo Metropolitan Rehabilitation Hospital (approval number: 2021–10). All participants were informed, both orally and in writing, about the content of the research, and a letter of informed consent was obtained from each participant before their enrolment. A participant in this study (Fig 2) has provided written informed consent (as outlined in PLoS consent form) for publication of images.

This study was conducted in a quiet laboratory environment in Tokyo Metropolitan Rehabilitation Hospital and Tokyo Metropolitan University. The period of study, including recruitment period was from 1 April 2021 to 31 March 2022.

The following criteria were established to exclude a possible reduced response to NMV: (1) healthy participants under 65 years of age [20,21]; (2) no pain at the site of vibration stimulation [22]; (3) no history of the spinal orthopedic disease [23]; and (4) no history of vestibular disease [9].

## Vibration settings

Vibratory stimulation was performed using a speaker-type stimulator [21,23] (Fig 1) set at a frequency of 80 Hz [5,6,12,14,18] and an amplitude of 0.8 mm [6,12]. The stimulation terminals were fixed with surgical tape and elastic lumbar bands to the upper trapezius muscle (10 cm medial to the acromion) on the left neck [12,13] and the multifidus muscle (3 cm lateral to the third lumbar spinous process) on the left lumbar back [6] (Fig 2).

## Outcome measures

Standing COP was measured using a COP platform (SR Vision by Sumitomo Riko Co. Ltd, Nagoya, Japan) to assess standing postural orientation. The frequency of signals was recorded at a sampling rate of 20 Hz to generate the COP data, which is valid and reliable for quantifying standing balance [24]. The participant was instructed to stand barefoot and upright on a COP platform with both hands by their sides [14]. Standing COP measurements were performed for 30 s each, starting with closed eyes and then with open eyes while gazing at landmarks placed at eye level 2 m away from the participant [12]. The percentage of weight-bearing on

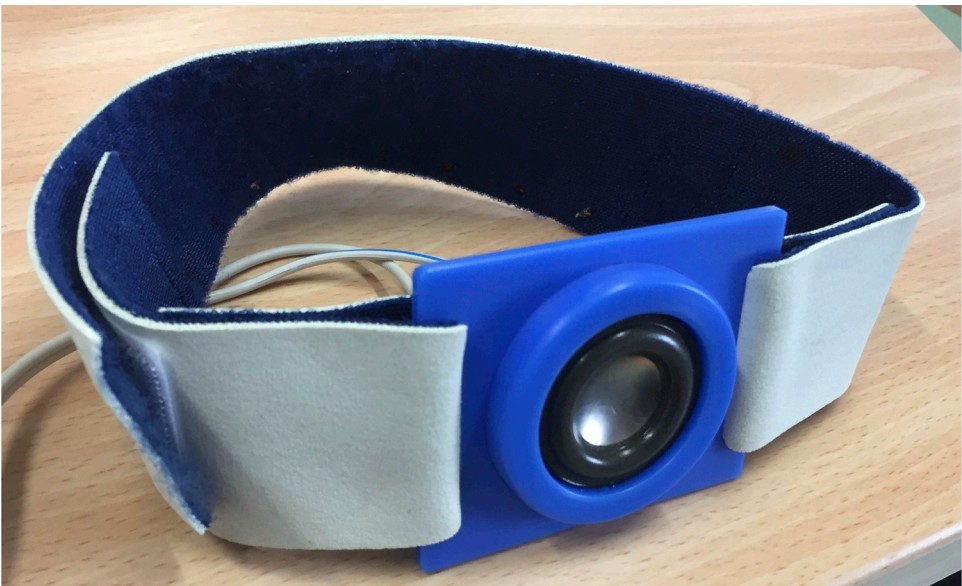

**Fig 1. Speaker-type stimulator.**

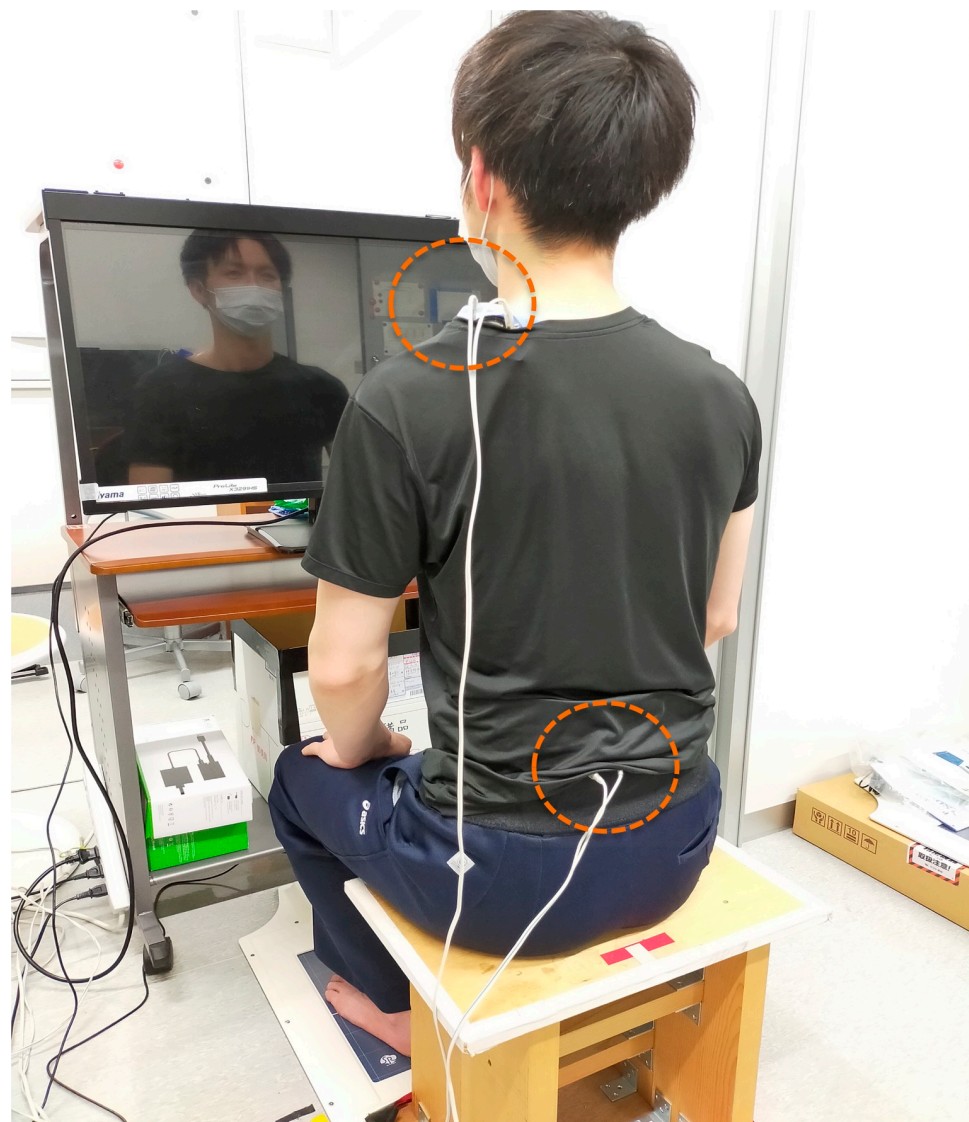

**Fig 2. Location of vibratory stimulators.** The vibratory stimulators are fixed in two circular positions in the sitting position.

the left limb (WBL) [%] and the mean mediolateral (ML) and anteroposterior (AP) position [mm], pathlength [mm], and surface area [mm$^2$] of COP were calculated. The pathlength of COP represents the total trajectory followed by the COP from its initial position to its final position. The surface area of COP represents a rectangle bounded by the maximum width of the sway in the ML and AP directions.

To assess spatial perception, SSA was measured using a 32" (1375 × 767 pixels) touch panel (NEWCOM Inc., Saitama, Japan) [25], which was placed in front of each participant. The position of the touch panel was adjusted such that when a participant reached it with their right index finger in an upright sitting position, the distance could be achieved with slight elbow flexion without excessive forward bending or trunk rotation. Participants were instructed to place their right index finger at the position of the manubrium of sternum in a closed eyes state. Subsequently, the participants were instructed to touch the location on the touch panel

with their index fingers, which they thought was directly in front of the manubrium of sternum. After two practice sessions with closed eyes, the exercise was repeated eight times, and the mean position in the ML and AP planes [mm] on the touch panel were calculated using Grid software (NEWCOM Inc., Saitama, Japan) [25]. During the practice sessions, we checked only the procedures and gave no feedback on the results.

## Procedures

The participants sat on a stable seat 40-cm high with both feet on the COP platform. Measurements were performed before and after stimulation. First, SSA was measured during the sitting position. Next, the participants stood up with their eyes open, and standing COP was measured with eyes closed for 30 s, and then with open eyes for 30 s. We tested four conditions: (1) no vibratory stimulation (control), (2) vibratory stimulation of the left neck for 30 s, (3) vibratory stimulation of the left neck for 10 min, and (4) vibratory stimulation of the left neck and left lumbar back for 10 min. Stimulation was performed with the eyes closed while in a resting, sitting position. Measurements in the four conditions were conducted with random cross-over using a random number table. There was a 5-min resting period between the conditions, and all procedures were conducted in approximately 90 minutes.

## Statistical analysis

Measurements of COP and SSA values were subtracted before and after each condition for standardized variation. Statistical analysis was conducted using one-way analysis of variance with repeated measures or Greenhouse–Geisser correction [26] when Mauchly's test of sphericity was significant [27] in all four conditions. When the results were significant, multiple comparisons were made using Dunnett's test. SPSS ver. 26.0 (IBM Corp., Armonk, NY, United States) was used to perform the statistical tests, and the significance level was set at 5%.

The sample size was calculated using G*Power version 3.1 [28], with an alpha level of 0.05 and a study power of 80%. The effect size was set at 0.25 with reference to Cohen's f [29]. Thus, the required sample comprised 24 participants.

## Results

The study included 24 participants (all right-handed; 12 males; 25.7±3.7 years; WBL with closed eyes 50.2±2.3%; WBL with open eyes: 50.4±2.8%), and they performed all four conditions, and a total of 96 trials was conducted.

A one-way analysis of variance with repeated measures showed that the left lower limb loading rate, ML-COP, AP-COP, and pathlength of COP in closed eyes were significant (Table 1). The results of Dunnett's test showed no significant differences in the loading rate of the left lower and pathlength of COP. ML-COP was significantly different between the control and simultaneous stimulation of the left neck and trunk (variation, 2.85; standard error, 1.05; p = 0.02; 95% confidence interval [CI], 0.33–5.37) (Fig 3A). AP-COP showed a significant difference between the control conditions and simultaneous stimulation of the left neck and trunk (variation, 5.21; standard error, 1.91; p = 0.02; 95% CI, 0.64–9.79) (Fig 3B).

## Discussion

This study aimed to clarify the effects of NMV on standing postural orientation and spatial perception in healthy participants, depending on differences in stimulation duration and simultaneous stimulation of trunk muscles. The results of this study showed that compared to the control condition, ML-COP was biased to the stimulus side and AP-COP was to the

**Table 1. The results of one-way analysis of variance with repeated measures.**

| Outcomes | Condition | Control | | Neck 30 s | | Neck 10 min | | Neck + Trunk 10 min | | Mauchly | | ANOVA | | |
|---|---|---|---|---|---|---|---|---|---|---|---|---|---|---|
| | | means | SE | means | SE | means | SE | means | SE | Mauchly's W | p value | F value | η2 | p value |
| WBL [%] | CE | -0.004 | 0.016 | -0.002 | 0.019 | 0.007 | 0.021 | 0.009 | 0.021 | 0.95 | 0.95 | 2.922 | 0.11 | 0.04[a] |
| | OE | 0.003 | 0.020 | -0.001 | 0.025 | 0.005 | 0.015 | -0.002 | 0.021 | 0.73 | 0.22 | 0.38 | 0.02 | 0.76 |
| ML-COP [mm] | CE | 0.78 | 3.07 | 0.3 | 3.88 | -0.98 | 3.94 | -2.07 | 3.68 | 0.87 | 0.76 | 3.19 | 0.12 | 0.03[a] |
| | OE | 0.46 | 4.1 | -0.39 | 3.86 | 0.17 | 4.45 | -1.1 | 3.24 | 0.75 | 0.29 | 0.67 | 0.03 | 0.57 |
| AP-COP [mm] | CE | -0.38 | 7.83 | -1.32 | 5.18 | 0.96 | 6.74 | 4.84 | 6.5 | 0.5 | 0.01[a] | 3.81 | 0.14 | 0.02[b] |
| | OE | -0.58 | 5.59 | -0.52 | 5.75 | 1.25 | 6.02 | 2.2 | 5.67 | 0.75 | 0.28 | 1.39 | 0.06 | 0.25 |
| Pathlength [mm] | CE | 5.15 | 12.26 | -10.76 | 29.33 | 7.97 | 41.65 | 15.69 | 44.47 | 0.8 | 0.43 | 2.87 | 0.11 | 0.04[a] |
| | OE | -5.29 | 25.76 | 5.74 | 28.27 | 6.26 | 31.58 | 4.15 | 24.2 | 0.95 | 0.96 | 0.87 | 0.04 | 0.46 |
| Surface area [mm²] | CE | -6.96 | 48.1 | 0.46 | 26.76 | 10.67 | 51.7 | 5.7 | 58.56 | 0.79 | 0.41 | 0.61 | 0.03 | 0.61 |
| | OE | 1.96 | 16.76 | 15.54 | 25.75 | 13.08 | 43.1 | -0.45 | 13.78 | 0.5 | 0.01[a] | 2.3 | 0.09 | 0.11 |
| ML-SSA [mm] | - | 5.79 | 78.23 | -11.3 | 77.74 | 6.63 | 89.22 | 20.33 | 115.22 | 0.8 | 0.45 | 0.48 | 0.02 | 0.7 |
| AP-SSA [mm] | - | 5.79 | 93.87 | -13.95 | 75.06 | -23.23 | 87.3 | -28.16 | 109.24 | 0.74 | 0.26 | 0.67 | 0.03 | 0.57 |

WBL, percentage of weight on the left limb; ML, mediolateral; AP, anteroposterior; CE, closed eyes; OE, open eyes: COP, center of pressure; SSA, subjective straight ahead

a, $p < 0.05$

b, Greenhouse–Geisser $p < 0.05$.

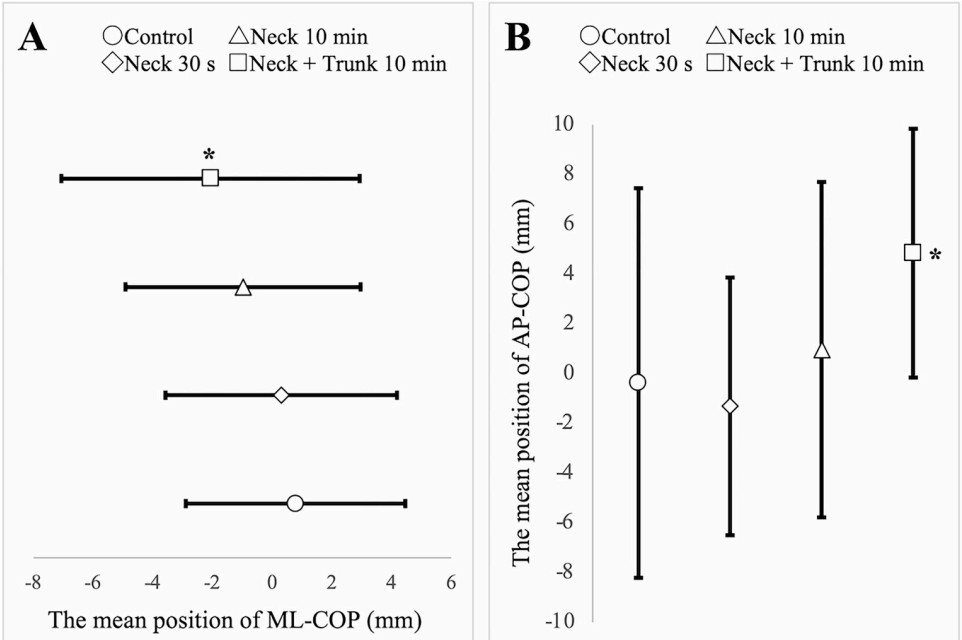

**Fig 3. Differences in the mean position of COP before and after vibration (n = 24).** Error bars indicate the standard error. A. The results of Dunnett's test on the mean position of ML-COP with closed eyes. Positive values indicate a rightward deviation, whereas negative values indicate a leftward deviation on the mediolateral plane. B. The results of Dunnett's test on the mean position of AP-COP with closed eyes. Positive values indicate an anterior deviation, whereas negative values indicate a posterior deviation on the anteroposterior plane. Asterisks indicate significant differences compared with the control condition, Dunnett's test: $p < 0.05$. ML-COP, mediolateral-center of pressure; AP-COP, anteroposterior-center of pressure.

anterior side immediately following NMV and trunk muscle vibration for 10 min. There were no significant differences between 30 s and 10 min of NMV with the control condition, and SSA showed no significant differences. Thus, the results suggest that combining NMV with vibratory stimulation of the trunk muscles intensifies the effect on the standing postural orientation.

Proprioceptive sensations from cervical muscles contribute to body representations, including the position and hierarchical arrangement of limbs and the organization of segments in space [9]. Continuing focal sensory input to Ia nerve fibers in muscle spindles induces the illusion of vibrated muscles being stretched even though they are not actually moving [5]. NMV induces the illusion that the cervical muscles are stretched, indicating that the standing position is tilted. The standing COP bias is considered as the response of CNS to this illusion and is the result of a modulation in the integration of the proprioceptive sensations [30]. The proprioceptive sensations transmitted by type Ia nerve fibers in muscle spindles rapidly communicate the changes in muscle length to the CNS. The broad hierarchy of the CNS composed of the vestibular nucleus [31], cerebellar nucleus ventricle [32], motor cortex [33], temporoparietal junction [34], subcortex [35], and cortex [36] has been implicated in the integration of proprioceptive sensations. However, the mechanism of the effect of NMV on the CNS is still unclear [9,10].

In healthy participants, the trajectory of gait is modulated by vibration stimulation of the trunk muscles [37], and motion perception is enhanced when the head is kept facing the side of trunk rotation [38], suggesting that proprioceptive sensation of the trunk may influence body representations. In this study, combining NMV and vibratory stimulation of the ipsilateral trunk muscles resulted in a significant deviation of the ML-COP and AP-COP to the stimulus and anterior sides, respectively, compared to the control condition. This suggests that the NMV-induced nervous system modulation would be amplified by proprioceptive sensory input to the trunk muscles. Bilateral cervical vibration stimulation produces anterior body tilt in the sagittal plane [5,30,39]. Vibratory stimulation of the upper trapezius muscle and the lumbar region of the trunk, which were the stimulation sites in this study, causes a bias in the standing COP, not only to leftward deviation on the mediolateral plane but also to anterior deviation on the anteroposterior plane [6]. We believe that simultaneous stimulation synthesized the direction of standing COP deflection and caused significant anterior bias.

SSA is ipsilateral to the stimulus after the end of vibratory stimulation in healthy participants [17] and patients with stroke and unilateral neglect [40–42]. SSA represents the relative position of egocentric body coordinates, such as the midline and sagittal axis of the body, to those in external space. Although several pointing techniques have been used in previous studies, each shows different results depending on whether they are referred to via vision with eyes open or proprioceptive sensation with closed eyes for their determination [15]. In the previous studies with visual SSA task, it is common to use a task in which the participant moves to a position that they judge to be the midline of a randomly presented luminous point or rod on a horizontal plane [17,43]. In the previous studies with proprioceptive SSA task, the participant's forearm was placed at a random angle on a rotating disc and the participant was asked to passively hold the rotating forearm at the midline point [15,44]. The former was performed in a dim room with the eyes open, whereas the latter was measured with closed eyes. NMV produces an illusion that the viewpoint presented in a dark room is biased in the opposite direction of the stimulus [4]. Participants sensitive to this visual illusion tend to have biased SSA [45,46]. Previous studies have reported a deviation in SSA by NMV using visual SSA [17,40–42]. In contrast, in another study that found a significant deviation in standing COP after 14 days of repeated NMV, there was no deviation in proprioceptive SSA [18]. This is consistent with the result of this study, suggesting that proprioceptive SSA does not deviate under the

measurement condition without visual illusion. It is also possible that SSA measurements in a stable sitting position, where sensory information is available from a wide base of support, did not lead to modulation of body representation by local sensory input upon simultaneous stimulation with NMV and trunk muscles. In future studies, the effect of simultaneous trunk stimulation on spatial cognition would be more apparent if both SSAs were included in the assessment measures.

This study was conducted in young individuals, mainly in their 20s, which is a limitation of this study. Whether similar results occur in other age groups is unclear. The effects of sex differences have also not been taken into consideration.

In conclusion, the findings suggest that proprioceptive input to the trunk muscles would amplify the central nervous system's modulator effect of NMV in the standing position. It is also possible that SSA based on proprioceptive sensation is not biased under closed-eye measurement conditions where visual illusions are not produced. Combining NMV with vibratory stimulation of the trunk muscles may be a new method for intensifying an effect in the standing postural orientation.

## Author Contributions

**Conceptualization:** Masaki Hirosawa, Itaru Takehara, Kazu Amimoto.

**Data curation:** Masaki Hirosawa, Yoshinao Moriyama, Kazu Amimoto.

**Formal analysis:** Masaki Hirosawa, Yoshinao Moriyama, Kazu Amimoto.

**Investigation:** Masaki Hirosawa, Kazu Amimoto.

**Methodology:** Masaki Hirosawa, Yoshinao Moriyama, Kazu Amimoto.

**Project administration:** Itaru Takehara, Kazu Amimoto.

**Resources:** Kazu Amimoto.

**Software:** Kazu Amimoto.

**Supervision:** Itaru Takehara, Kazu Amimoto.

**Validation:** Yoshinao Moriyama, Kazu Amimoto.

**Visualization:** Yoshinao Moriyama, Kazu Amimoto.

**Writing – original draft:** Masaki Hirosawa, Kazu Amimoto.

**Writing – review & editing:** Masaki Hirosawa, Itaru Takehara, Yoshinao Moriyama, Kazu Amimoto.

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
