## [Decision Letter · Decision Letter 0]

28 Nov 2022

PONE-D-22-21542Effects of unilateral neck muscle vibration on standing postural orientation and spatial perception in healthy subjects based on stimulus duration and simultaneous stimulation of trunk musclesPLOS ONE

Dear Dr. Amimoto,

Thank you for submitting your manuscript to PLOS ONE. After careful consideration, we feel that it has merit but does not fully meet PLOS ONE’s publication criteria as it currently stands. Therefore, we invite you to submit a revised version of the manuscript that addresses the points raised during the review process.

We look forward to receiving your revised manuscript.

Kind regards,

Tomoyoshi Komiyama, Ph.D

Academic Editor

PLOS ONE

Journal Requirements

2. We note that Figure 2 includes an image of a participant in the study. 

Additional Editor Comments:

Dear Authors,

Your study aimed to clarify the effects of NMV on standing postural orientation and spatial perception in the same healthy participants, depending on differences in stimulation duration and simultaneous stimulation of trunk muscles. Your findings suggest that proprioceptive input to the trunk muscles amplifies the central nervous system’s modulator effect of NMV in the standing position.

However, I think you should strengthen the reliability of these results by adding as much information as possible. We thus have some questions and suggestions for the manuscript that you might consider. I believe these comments will be very helpful in the revision of your study.

For example, I have some concerns regarding your findings.

It is questionable whether difference in age and gender of the subjects is reflected in your results.

Two reviews below are from reviewers with feedback for your manuscript.

I have carefully considered your manuscript, however, I decided to major revision your study based on these comments.

Please consider this feedback in the revision of your manuscript.

Tomoyoshi Komiyama

Reviewers' comments:

Reviewer's Responses to Questions

**Comments to the Author**

1. Is the manuscript technically sound, and do the data support the conclusions?

Reviewer #1: Yes

Reviewer #2: Partly

2. Has the statistical analysis been performed appropriately and rigorously? 

Reviewer #1: Yes

Reviewer #2: Yes

3. Have the authors made all data underlying the findings in their manuscript fully available?

Reviewer #1: Yes

Reviewer #2: Yes

4. Is the manuscript presented in an intelligible fashion and written in standard English?

Reviewer #1: Yes

Reviewer #2: No

5. Review Comments to the Author

Reviewer #1: This study examined the effects of NMV on standing posture orientation and found that intrinsic sensory input to the trunk muscles amplified the effects of NMV in the standing posture. This may be a new way to combine NMV with vibration stimulation of the trunk muscles to strengthen the effects of standing posture holding.

The following points need to be clarified.

1 In L170, the study subjects are listed as 24 subjects, but since this is a crossover study, the actual number of subjects should be 12.

2 It does not state why the study was limited to males in their 20s as subjects. In the discussion, there is no mention of age- and sex-related differences in balance ability among adolescents. A prior paper, Age and sex differences in human balance performance from 6-18 years of age: A systematic review and meta-analysis. PLoS One. 2019 Apr 9;14(4):e 0214434. shown, and the entire experimental design should be restructured. Regarding postural control, we believe that brain development should also be considered.

Reviewer #2: The authors of this study were interested in investigating the effects of neck muscle vibration and lumbar muscle vibration over different time periods and in isolation or combination. Outcomes measures of interest were percentage of weight-bearing by the left leg, AP and ML CoP position, length, and surface area of CoP, as well as measurement of subjective straight ahead utilizing an electronic touch board. Overall, the study appears to be well conducted, though some of the descriptions of techniques and measures made interpretation difficult.

Line 22-24: The sentence beginning with “However, the modulating effects….” is difficult to follow. I am unclear on what exactly is being modulated as well as what exactly is unknown – please rephrase this sentence to be more clear (or perhaps separate into two, simpler sentences).

Line 48: The use of “almost selectively” here is unclear. Do you mean that focal muscle vibration is not the only activator of primary nerve endings? Or that focal muscle vibration mostly activates primary nerve endings, but also affects other structures? Please clarify.

Line 62: Please change “a previous study” to “previous studies” as you then cite two studies.

Line 75: This sentence was a little difficult to follow; I would recommend rephrasing to clarify that SSA deviates towards the stimulus side during and after NMV, in both healthy and stroke populations (if this is indeed the correct reading). As it is currently, it can be interpreted as “those having a stroke during and after NMV” (as in, those who suffered a stroke during vibration).

Line 82: I believe this sentence is explaining that vibratory stimulus of the neck and lower limbs leads to CoP deflection to the opposite side of stimulus, and vibration of the neck and trunk leads the CoP to the same side as the stimulus. Otherwise, if it is specifying the direction of CoP deflection from those sites individually, the neck would be responsible for both ipsilateral and contralateral deflection. If the first option is true, please make the sentence more explicit about this; if the second option is true, please explain the details of neck vibration causing CoP deflection in both directions.

Line 84: The authors state “the direction of standing CoP deflection is similar,” but it is not clear what this direction is similar to. Please clarify.

Line 107: This is minor, but please remove “the”, as you are referencing any vestibular disease, rather than a particular one.

Line 111: I very much appreciate the inclusion of the frequency and amplitude parameters of vibration, though it would be helpful to readers to include a rationale for these choices (particularly for the amplitude).

Line 112: The authors reference “anatomical positions” and cite work by Courtine et al; however, this paper makes no explicit reference to any anatomical positions. Please clarify.

Line 129: Participants, upon opening their eyes, gazed at “landmarks” – does this refer to the marker placed at eye level on line 126?

Line 131: I was unclear on what ‘CoP length’ was referring to. On line 174, the authors refer to “total trajectory length,” which implies – if this refers to the same measure – that it could be pathlength. However, this was not clear from the methods. Either a short description, or a citation would help readers to follow this metric.

Line 138: I am unclear what “sternal pattern” is referring to in this sentence. Was the touch panel projecting an image of their body that they reached for? Or is this just referring to the sternum of each participant?

Line 140: The participants were given two practice sessions; were they provided any feedback on their practice (or any other) attempts?

Line 147: Were participants allowed to open their eyes to stand? Or did their eyes remain closed the entire time? Similarly, when did data collection start for the eyes open and closed periods?

Line 206: Minor grammar - please change “was no significant difference” to “were no significant differences”

Line 214: “The integration of proprioceptive sensations by NMV” implies that NMV is actually the entity/thing performing the integration. Do the authors mean the influence of NMV on proprioception, perhaps? Please clarify.

Line 215: It is not clear what is meant by “implicated.” Please clarify.

Line 218: The authors state that focal sensory input from NMV alters body representations by modulating the effects of the central nervous system. The preceding lines imply that this may be from NMV exerting some influence on Ia afferents; however, this is never explicitly stated. Please consider a linking sentence to clarify both what “effects of the CNS” represent as well as how - specifically - focal vibration is modulating them.

Line 226: The authors state that NMV-induced nervous system modulation is amplified by proprioceptive sensory input to the trunk; however, in this study, vibration of the neck alone did not actually have any significant effects. Is this amplification based on previous literature (in which case it would help to state such), or is the nature of the change less amplification and more attributable to the combination?

Line 230: It is not clear what is meant by “anterior forehead” and what “component” is being referred to. Please clarify.

Line 241: It would benefit a reader to clarify that this is in reference to a different study, and not the current one.

Line 257: This reflects my comments on lines 218 and 226. Please address those comments and update here as appropriate.

6. PLOS authors have the option to publish the peer review history of their article (what does this mean?). If published, this will include your full peer review and any attached files.

Reviewer #1: No

Reviewer #2: No

---

## [Author Response · Author response to Decision Letter 0]

11 Dec 2022

Tomoyoshi Komiyama

Academic-Editor

PLoS One

Dear Editor: 

We wish to re-submit the manuscript titled “Effects of unilateral neck muscle vibration on standing postural orientation and spatial perception in healthy subjects based on stimulus duration and simultaneous stimulation of trunk muscles.” The manuscript ID is PONE-D-22-21542.

We thank you and the reviewers for your thoughtful suggestions and insights. The manuscript has benefited from these insightful suggestions. I look forward to working with you and the reviewers to move this manuscript closer to publication in the PLoS One.

The manuscript has been rechecked and the necessary changes have been made in accordance with the reviewers’ suggestions. The responses to all comments have been prepared and attached herewith.

Thank you for your consideration. I look forward to hearing from you.

Sincerely,

Kazu Amimoto

Department of Physical Therapy, Faculty of Human Health Sciences

Tokyo Metropolitan University, Tokyo, Japan

Tel: +81 3-3819-1211

Email: amimoto@tmu.ac.jp

---

## [Decision Letter · Decision Letter 1]

13 Jan 2023

Effects of unilateral neck muscle vibration on standing postural orientation and spatial perception in healthy subjects based on stimulus duration and simultaneous stimulation of trunk muscles

PONE-D-22-21542R1

Dear Dr. Amimoto,

We’re pleased to inform you that your manuscript has been judged scientifically suitable for publication and will be formally accepted for publication once it meets all outstanding technical requirements.

Kind regards,

Tomoyoshi Komiyama, Ph.D

Academic Editor

PLOS ONE

Additional Editor Comments (optional):

Dear authors,

Thank you for submitting your revised manuscript.

It was much easier to understand than the original manuscript.

You have satisfactorily addressed the comments from the two reviewers.

Therefore, I have no further suggestions. I am satisfied with the responses and the edits, so I am happy to accept you study.

Also, I believe this manuscript will satiate the reader's interest.

Tomoyoshi Komiyama

Reviewers' comments:

Reviewer's Responses to Questions

**Comments to the Author**

1. If the authors have adequately addressed your comments raised in a previous round of review and you feel that this manuscript is now acceptable for publication, you may indicate that here to bypass the “Comments to the Author” section, enter your conflict of interest statement in the “Confidential to Editor” section, and submit your "Accept" recommendation.

Reviewer #1: All comments have been addressed

Reviewer #2: All comments have been addressed

2. Is the manuscript technically sound, and do the data support the conclusions?

Reviewer #1: Yes

Reviewer #2: (No Response)

3. Has the statistical analysis been performed appropriately and rigorously? 

Reviewer #1: Yes

Reviewer #2: (No Response)

4. Have the authors made all data underlying the findings in their manuscript fully available?

Reviewer #1: Yes

Reviewer #2: (No Response)

5. Is the manuscript presented in an intelligible fashion and written in standard English?

Reviewer #1: Yes

Reviewer #2: (No Response)

6. Review Comments to the Author

Reviewer #1: The authors have published the necessary information on the points raised and have provided an informed and appropriate response. In applying the methodology of this study as an evaluation measure in future clinical trials, the authors are to be commended for indicating the research limitations in their discussion.

Reviewer #2: (No Response)

7. PLOS authors have the option to publish the peer review history of their article (what does this mean?). If published, this will include your full peer review and any attached files.

Reviewer #1: No

Reviewer #2: No
